# Automated 1D Helmholtz coil design for cell biology: Weak magnetic fields alter cytoskeleton dynamics

Abasalt Bahrami[1,2], Leonardo Y. Tanaka[2], Ricardo C. Massucatto[2], Francisco R. M. Laurindo[2], Clarice D. Aiello[3]*

1 Department of Electrical and Computer Engineering, University of California, Los Angeles, California, United States of America, 2 Vascular Biology Laboratory, Heart Institute (InCor), University of São Paulo School of Medicine, São Paulo, Brazil, 3 Quantum Biology Institute, Los Angeles, California, United States of America

* clarice@quantumbiology.org

**Data availability statement:** All data is available in GitHub at the following repository: https://github.com/QuBiTUCLA/biocoil.

## Abstract

Evidence of the biological impacts of weak magnetic fields have been reported for more than fifty years. However, research progress on such effects has been hampered by a lack of systematics in most experiments. Efforts to increase the systematics in such cell biology experiments must include the capability of producing fields that can be automatically adjusted and that are stable throughout an experiment's duration, usually operating inside an incubator. Here, we report on the design of a fully automated 1D Helmholtz coil setup that is internally water cooled, thus eliminating any confounding effects caused by temperature fluctuations. The coils also allow cells to be exposed to magnetic fields from multiple directions through automated controlled rotation. Preliminary data, acquired with the coils placed inside an incubator and on a rat vascular smooth muscle cell line, confirm previous reports that both microtubule and actin polymerization and dynamics are altered by weak magnetic fields.

## Introduction

The scientific community has long been intrigued by the effects of magnetic fields (MFs) on biology across the tree of life, as reported by numerous studies [1–5]. Despite extensive research spanning more than half a century, the mechanisms underlying the observed effects on cellular systems remain unclear and ambiguous. In some instances, the effects themselves are contentious or difficult to replicate. Even though cell biology also seem to respond to both strong fields and MF gradients, in this work we focus on the biological effects of uniform and *weak* MFs [6–8]. By *weak*, we mean fields whose intensity are up to approximately ten times that of the Earth's MF, which is around 25–65 $\mu$T.

Weak MFs impact a myriad of cellular processes, including the regulation of gene expression [9], ion channel functioning [10], and the production of reactive oxygen species [11–15]. MFs can also impact apoptosis in both cancer and normal cells [16–21]. MFs influence metabolic pathways [22], inducing changes in cell growth rate [23,24], affecting cellular morphology [25–28], and promoting muscle cell development [29]. Finally, it is important to

**Funding:** This work was supported by: Young Investigator-FAPESP (Fundação de Amparo à Pesquisa do Estado de São Paulo) grant 2018/07230-5, scholarship grants 2020/04280-1, 2022/16629-4, and 2023/05600-8 to L.Y.T.; CEPID (Centro de Pesquisa Inovação e Difusão)/FAPESP grant 2013/07937-8 to F.R.M.L.; Pioneer Science-Instituto D'Or grant, ONR (Office of Naval Research) grant N00014-21-1-4011, NSF (National Science Foundation)-Nano EAGER grant 2041158, and NSF-Bio EAGER grant 2114144 to C.D.A. The funders had no role in study design, data collection and analysis, decision to publish, or preparation of the manuscript. publish, or preparation of the manuscript.

**Competing interests:** The authors have declared that no competing interests exist.

recognize that MF effects do not necessarily monotonically increase with the strength of the applied field: *higher magnetic field intensities do not necessarily lead to more pronounced effects* [6,8].

Understanding the effects of being exposed to small-magnitude fields such as those produced by the low-level electromagnetic pollution that permeates our environment is a timely endeavor. Moreover, such an electromagnetic 'knob' in biology could be, and has already been, rationally used for medical theranostics [30,31], with magnetic therapies proposed to accelerate diabetic wound repair [32] and pain management [33], and to counter neurodegenerative diseases [34,35] and cancer [36–39].

Well-controlled and reproducible experiments are crucial to unambiguously establish and understand the biological effects of weak MFs. Certain bioelectromagnetics experiments employ permanent magnets, which present challenges like fixed field strengths and uneven magnetic fields. In contrast, coil setups, such as those obeying a Helmholtz configuration, offer the advantage of controllably changing uniform MFs [40–42]. Nevertheless, most current designs limit the MF application to a single direction, requiring manual coil rotation for direction changes. Our new design of 1D Helmholtz coils for bioelectromagnetic experiments overcomes the constraints of traditional setups by introducing complete automation of the MF direction. Additionally, we incorporate an effective cooling mechanism to mitigate temperature rise effects on cells. Many currently employed coils in bioelectromagnetic experiments lack an active cooling system, posing challenges for conducting prolonged experiments under sustained MFs [43–45] and maintaining stable temperature conditions.

## Coil design

We designed a set of Helmholtz coils with water cooling. Unlike conventional coils that often, during operation, produce temperature fluctuations [46], this design efficiently prevents temperature rise by allowing for the regulation of the water flow rate (SF1 Fig). This set of coils can be controllably rotated around its pivot using a servo motor. We could not detect any MFs generated by the servo motor within our measurement accuracy of $(1.0 \pm 0.1)$ $\mu$T. By allowing the coil assembly to rotate, biological samples could be subjected to uniform 1D MFs at various orientations.

We chose a design capable of producing MF as strong as 15 mT, with this upper limit being determined by the size of the copper wire thickness 0.71 mm (22 gauge wire) chosen to wound the coils. Magnetic fields were numerically simulated using the Biot-Savart law. The axial and radial MFs produced by each coil in the Helmholtz configuration are mathematically given by [47]:

$$B_a = \frac{B_0}{\pi\sqrt{Q}}\left[E(k)\frac{1-\alpha^2-\beta^2}{Q-4\alpha} + K(k)\right], \qquad (1)$$

$$B_r = \frac{B_0\gamma}{\pi\sqrt{Q}}\left[E(k)\frac{1+\alpha^2+\beta^2}{Q-4\alpha} - K(k)\right], \qquad (2)$$

where:

$B_a$ is the field component aligned with the axis [G],

$B_r$ is the field component in a radial direction [G],

$a_0$ is the loop radius [m],

$\mu_0$ is the permeability constant $[4\pi \times 10^{-7}\,\mathrm{N}\cdot\mathrm{A}^{-2}]$,

$x$ is the axial distance from the center of the coil [m],

$r$ is the radial distance from the axis of the coil [m],

$\alpha = r/a_0, \quad \beta = x/a_0, \quad \gamma = x/r,$

$Q = (1 + \alpha)^2 + \beta^2,$

$k = 4\alpha\sqrt{Q},$

$B_0 = i\mu_0 \dfrac{2a_0}{\sqrt{Q}}$ is the magnetic field at the center of the coil [G],

$i$ is the current in the loop wire [A],

$K(k)$ is the complete elliptic integral function of the first kind, and

$E(k)$ is the complete elliptic integral function of the second kind.

Using the formulas presented in Eq 1 and Eq 2, we determined the coil dimensions required to produce an axial MF with a maximum chosen strength of 15 mT. Taking into account several factors, including the desired dimension of our sample (we optimized the design for cells hosted within a 35 mm Petri dish), the size of a typical cell incubator, MF homogeneity and our available water system, we chose the coil dimensions as listed in Table 1.

The enclosure of the copper coils is 3D-printed using thermoplastic material, which offers the advantages of being lightweight, corrosion-free, and of not experiencing air condensation at lower temperatures, as opposed to metals. A total of 96 turns of copper wires, each with a diameter of 0.71 mm, were wound around the coil holder and we apply thermally conductive glue on the surfaces of the coils. The wound coils are placed within the coil enclosure, and both electrical and water connections are established. A two-part epoxy is employed to seal the electrical connections. To prevent water leaks, we apply silicone sealant and firmly fixed the caps to the coil enclosure before screwing them. Fig 1 depicts an exploded cross-section view to scale.

Fig 2, left, features the computer-assisted design of the set of coils, highlighting its various components. Fig 2, right, shows a photo of the design; note the integration of a servo motor to rotate the coil. Additionally, a thermocouple wire has been incorporated in a holder beneath the sample Petri dish for real-time monitoring of temperature fluctuations. Following the assembly of the coils, we did water-leak tests and measured the produced MF when different currents were applied. A 3-axes magnetometer is used to measure the produced MF.

**Table 1. Geometrical details of the designed coil.**

| Parameter | Symbol | Value |
|---|---|---|
| Inner diameter of enclosure (mm) | $D_i$ | 54 |
| Outer diameter of enclosure (mm) | $D_o$ | 114 |
| Width of enclosure (mm) | $w$ | 20 |
| Copper wire thickness (mm) | $t$ | 0.71 |
| Inner diameter of coil holder (mm) | $d_{coil}$ | 80 |
| Width of coil holder (mm) | $w_{coil}$ | 16 |
| Number of turns in each coil | $N$ | 96 |
| Resistance of coil ($\Omega$) | $R$ | $2.74 \pm 0.01$ |
| Voltage drop in coil (V) | $\Delta V$ | $2.75 \pm 0.01$ |
| Coil inductance ($\mu$H) | $L$ | $870 \pm 15.0$ |
| Inner diameter of water tube (mm) | $d_i$ | 4 |
| Outer diameter of water tube (mm) | $d_o$ | 6 |
| Length of water tube (m) | $l$ | 4 |

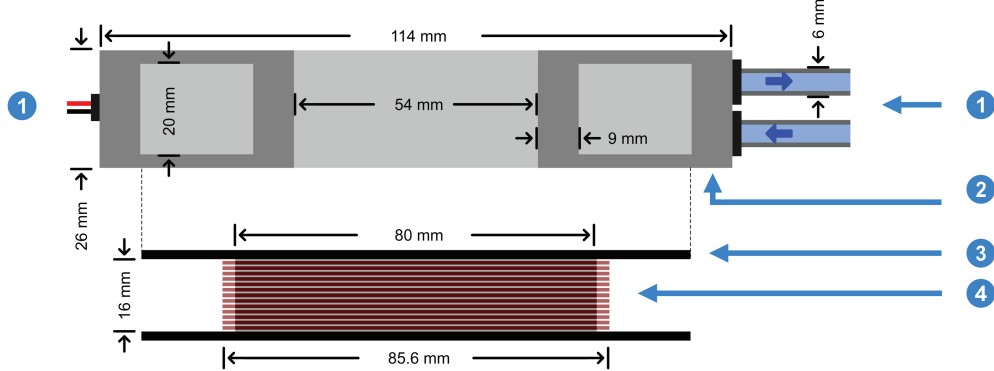

**Fig 1. Exploded cross-sectional view of a single coil in the designed holder with detailed dimensions**. It includes the water and electrical connections (1), the enclosure for the coil made of Delrin material (2), the 3D-printed coil holder (3) and the wound copper coil with 96 turns of copper (4). The coil fits within the enclosure.

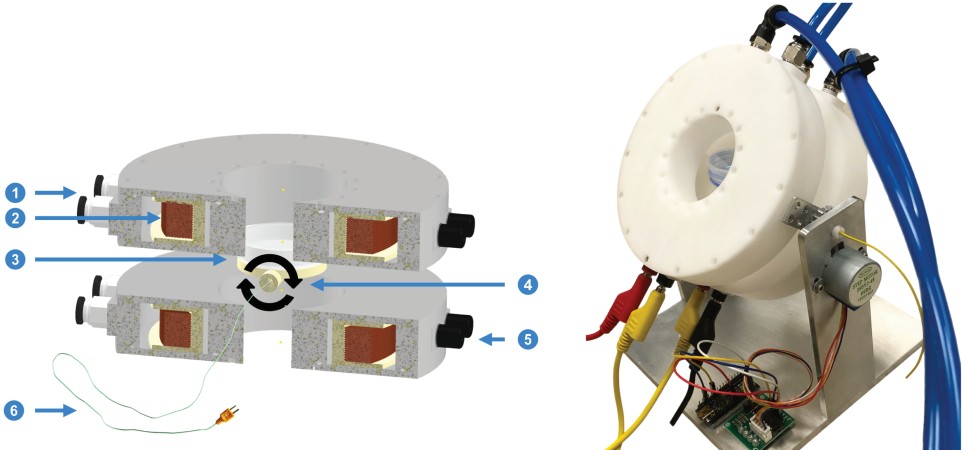

**Fig 2. Computer-assisted design and photo of the water-cooled, 1D Helmholtz coil set.** Left, components of the designed coil, namely: (1) water cooling connections; (2) copper coil contained within the enclosure; (3) holder for 35 mm Petri dish; (4) a pivot point demonstrating the swinging motion of the coils while the Petri dish remains stationary; (5) electrical connections; and (6) temperature sensor employed to accurately monitor the temperature directly below the Petri dish within the cell culture area (resolution of 0.1 °C). Right, fully assembled coil set.

**Detailed instructions for coil construction are freely available on our GitHub repository** [48].

## Field simulations and measurements

A Helmholtz configuration is achieved by placing the two coils in our arrangement at a distance of 43.4 mm, which is equal to the radius of each coil [49]. In order to confirm the simulated magnetic flux values and the extent to which the produced MFs are homogeneous, we created a numerical model in COMSOL Multiphysics [50]. Our simulated MF values are shown in Fig 3, for a current of 1 A.

When a simulated current of 1 A flows through the coils, the MF variation over a 35 mm Petri dish has a maximum axial (radial) divergence of 4.2 (1.7 $\mu$T. We performed MF measurements along the axial ($B_a$) and radial ($B_r$) directions, and the results closely match

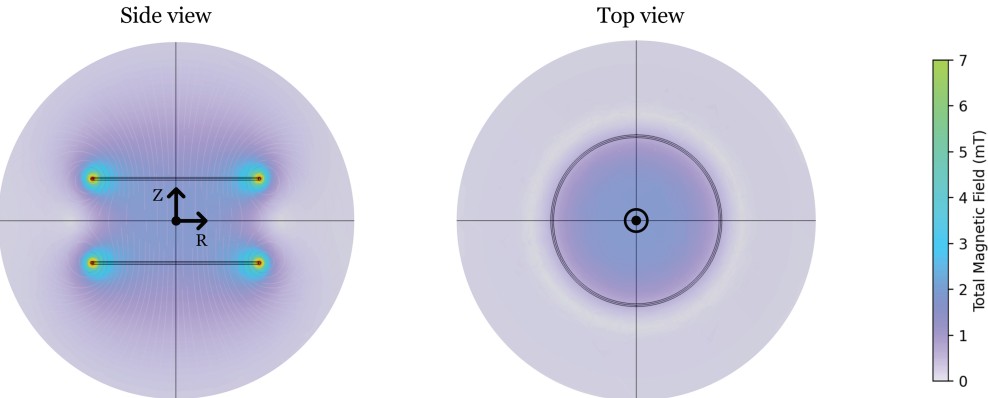

**Fig 3. COMSOL simulation depicting the uniform magnetic field generated by the designed coil set.** This simulation, for a current of 1 A, indicates that the MF within the coil is uniformly distributed, measuring $(1.80 \pm 0.01)$ mT over the 35 mm Petri dish. This uniform MF primarily results from the axial component, as the symmetric nature of the coil cancels out the radial magnetic components.

our simulations. At a current of $I = 1$ A, we measured $B_a = (1.850 \pm 0.015)$ mT and $B_r = (0.000 \pm 0.002)$ mT. The measurement error typically grows with the applied current due to imperfections in the coil winding process.

If changing the MF frequency is desired, it is crucial to acknowledge the physical limitations of producing alternating MFs. One of these limitations is due to the coils' inductance, which introduces a time delay in the current buildup process. The coils are measured to have inductance $L = 870\,\mu$H, resistance $R = 2.74\,\Omega$ and a voltage drop of $\Delta V = 2.75$ V. Although a voltage simulation indicates a rapid increase to 2.75 V within a few microseconds, due to the inductive reactance of the coil, achieving a current of 1 A and returning to zero, experimentally, takes about 130 ms when toggling the coil on and off. It is thus essential to ensure that the switching frequency does not exceed $f \sim 7$ Hz in this situation; otherwise, the current in the coils will be lower than expected. If there is a requirement to operate at higher frequencies, it becomes necessary to adjust the voltage accordingly to achieve the intended current (in other words, to perform impedance matching).

### Efficient heat dissipation in the coils

We analyzed the temperature increase of the water within a coil subjected to electric currents (see SSI Sect). The power dissipated by the coil contributes to heat transfer to water through convection. Experimentally, there is no observed temperature increase exceeding 0.1 °C when the coils operate at a current of 1 A, generating approximately $\sim 1.8$ mT. SF1 Fig illustrates the simulated temperature rise in the coils based on current and water flow rates, showcasing the system's performance under various conditions.

### Controlling moisture on coil surfaces

The dew point, denoting the temperature at which air becomes saturated with moisture and condensation initiates [51], needs to be considered in order to avoid water droplets to form onto the sample dish. Additionally, maintaining a higher dew point temperature can avert complications such as condensation-induced corrosion and potential electrical issues. In SSII Sect, we have analyzed the dew point temperature for the coils placed inside the incubator versus the incubator's own temperature and humidity. Under typical incubation parameters

(namely, temperature 37 °C and humidity 60%), we thus must set the water cooling temperature inside the coil enclosure to be above 28 °C to avoid moisture condensation (SF2 Fig).

## Weak magnetic fields effects in cytoskeleton dynamics

We deploy the coils to study the effects of weak MFs on the fundamental process of cell cytoskeleton assembly, which is central in distinct stages of genesis and in the resolution of many diseases. We use A7r5, a cell line of vascular smooth muscle cells (VSMCs) that is responsive to mechanical stimulation and that has been shown to regulate tensional home-ostasis by coupling local oxidant generation to actin cytoskeleton remodeling [52]. To the best of our knowledge, the question of whether cytoskeleton remodeling in VSMCs can be modulated by MF stimulation had not been previously addressed.

## Weak magnetic fields disturb cytoskeleton reassembly after nocodazole exposure

We hypothesized that a weak MF would affect cytoskeletal organization in conditions in which more intense reorganization takes place—*e.g.*, during development or tissue repair responses. To model these conditions, we exposed the VSMCs to a reversible microtubule (MT)-disrupting agent, nocodazole, and observed the dynamics of cytoskeletal recovery after nocodazole withdrawal.

Fig 4 depicts the experimental protocol, which starts with the seeding of approximately 10,000 A7r5 VSMCs in each well of a 8-well chamber (0.7 cm$^2$/well). After 24 h, the serum-rich (10%) Dulbecco's Modified Eagle Medium (DMEM) culture medium is replaced with serum-free DMEM solution within the wells for at least 1 h. Followed serum starvation, cells were treated with nocodazole 10 $\mu$M or vehicle (DMSO) during 30 min. To eliminate any residual substances, the cells were washed twice with phosphate buffered saline (PBS) before being supplied with fresh serum-free medium. Subsequently, the cells are subjected to a MF of 480 $\mu$T, in a direction perpendicular to the bottom of the 8-well chamber, for a duration of 3 hours. *The magnetic field strength and exposure duration were chosen guided solely by existing literature* [6]. After completing the protocol, the medium was removed and the cells were fixed in paraformaldehyde 4% (15 min at 37 °C). Subsequently, wells were washed with PBS, followed by permeabilization with Triton X-100 (0.1% in PBS) for 10 min at 37 °C. Once done, cells were washed with PBS and incubated with blocking solution (normal goat serum 10%) for 60 min at room temperature (RT). After removing the blocking solution, the cells were subjected to staining using anti-mouse-tubulin antibody (1:200) for 12 h at 4°C in a wet chamber. The primary tubulin antibody was washed twice with PBS; the secondary antibody for tubulin, anti-mouse 488 nm (1:200), as well as nuclear and F-actin markers DAPI and phalloidin 635 nm, respectively, were incubated during 60 min with protection from light and at RT. Finally, wells were washed twice with PBS and slides were mounted with PBS/glycerol (1:1, v/v).

After the nocodazole challenge, MFs were observed to affect the reorganization of both the MT network and the F-actin architecture, as shown in Fig 5.

The fluorescence images in Fig 5A are representative of 229 cells obtained over the course of two identical technical replicates; they were all taken with identical exposure parameters. They reveal that the MF alone, at least at this intensity (480 $\mu$T), direction (vertical axis) and time of exposure (3 h), does not affect the architectures of MTs and F-actin (third row in the figure; cells similar to the control cells shown in the first row). Moreover, MTs were recovered after nocodazole washout in control cells under the ambient geomagnetic field (second row in the figure), as expected, but not in cells that were exposed to the external MF (fourth

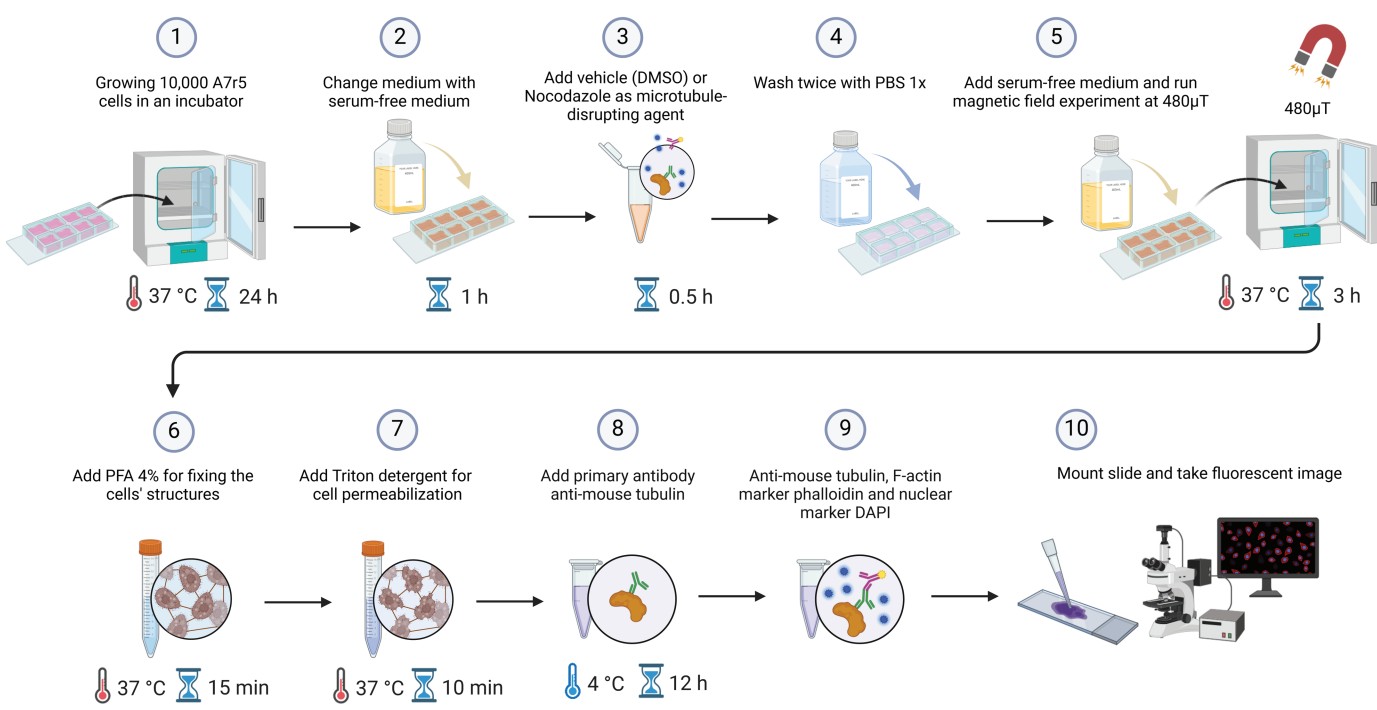

**Fig 4. Experimental procedure for examining cytoskeleton reassembly after nocodazole exposure.**

row in the figure). That is, after a short challenge with nocodazole, the MF induced tubulin accumulation at the cell periphery in patches disconnected with the core of the MTs. Given the strength of these effects and the primary effects of nocodazole in MTs, we propose that MTs are the focus of MF effects and that the loss of proper MT polymer formation and direction could be affected by the weak MF application. In parallel, application of the MF also affected the pattern of F-actin architecture during recovery after nocodazole: there are more pronounced stress fibers (seen as increased F-actin staining), with decreased fiber alignment. We then investigated in greater detail whether the loss of proper actin polymer formation and direction could also be affected by the weak MF application.

We quantified the degree of F-actin alignment using the Image J [53] plugin OrientationJ [54]. Fig 5B shows the calculated F-actin alignment level (a parameter varying from 0 to 100%) over the 229 whole cells, spanning the four experimental conditions (cells with or without nocodazole challenge; cells under the ambient geomagnetic field, or exposed to the external MF). Values for individual cells are indicated with a horizontal line; the violin shapes represent density curves whose width corresponds to the approximate frequency of data in each region. The violin shapes have been truncated at 0 as this is the minimum actin alignment level. The average value for each condition is indicated with a circle; the vertical bars indicate one standard deviation above and below the average value. The medians (which are of relevance as the data are not normally distributed) are indicated with a cross. Notably, the median alignment level for the nocodazole-disrupted cells under the ambient field (14.5%) is $\sim 1.6$ that of the nocodazole-disrupted cells under the applied MF (8.7%). The data not being normally distributed, we employ the Mann-Whitney statistical test to probe for a difference between those two populations, obtaining a p value of 0.0278; the star represents a $p \leq 0.05$

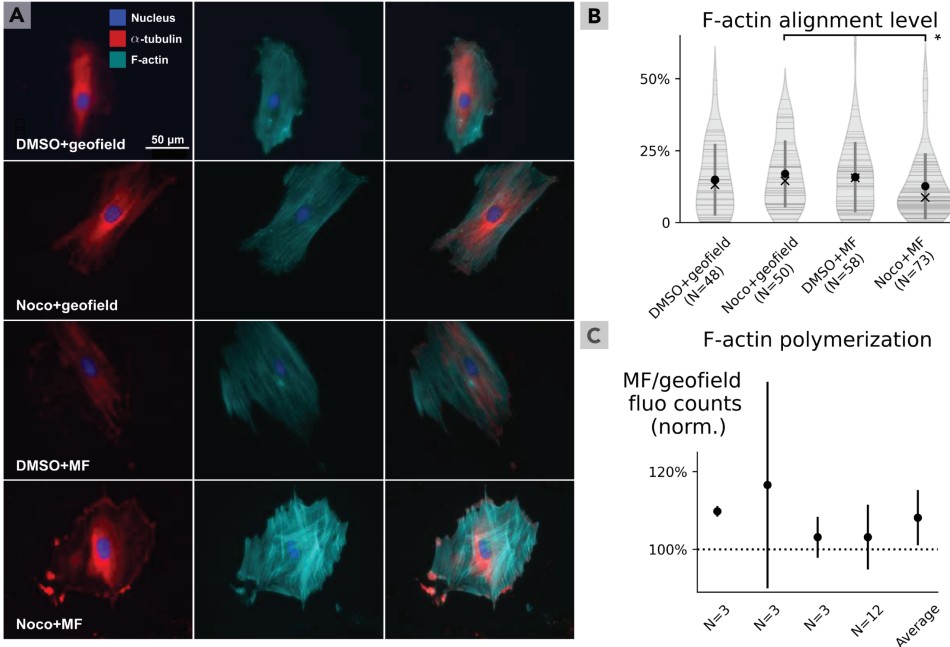

**Fig 5. Weak magnetic fields disturb cytoskeleton reassembly after nocodazole exposure, and influence actin polymerization. (A) Cells exposed to the weak magnetic field exhibit impaired microtubule recovery, F-actin fibers that are less aligned but higher in number, and a lack of mature focal adhesions.** Representative images out of 229 cells, obtained over the course of two identical experiments. Microtubule ($\alpha$-tubulin shown in red, leftmost panels) recovery was evaluated: in the absence ('DMSO') or after a 30 min disruption with nocodazole ('Noco'); under the ambient magnetic field ('geofield') or during exposure to a vertical 480 $\mu$T magnetic field for 3 h ('MF'). Actin cytoskeleton filaments (F-actin shown in green, middle panels) were altered by the weak magnetic field. Nucleus staining (in blue) is depicted in all panels. The rightmost panels depict merged microtubule, actin and nuclei stainings. **(B) The weak magnetic field decreases F-actin alignment during recovery after nocodazole.** The F-actin alignment (between 0 and 100%) was quantified over 229 whole-cell images. The median alignment level for the nocodazole-disrupted cells under the ambient field (14.5%) is ∼ 1.6 that of the nocodazole-disrupted cells under the applied magnetic field (8.7%), with a p = 0.0278 ≤ 0.05 indicated with a star. **(C) F-actin polymerization is slightly increased in the presence of a weak magnetic field.** Four independent plate reader-based fluorescence assays were performed in 2N (with N = {3,3,3,12}) individual wells, with fluorescence intensity being a proxy for actin polymerization levels. After 1 h of exposure to a vertical 480 $\mu$T magnetic field, the exposed cells emit ∼ 8% more fluorescence than the cells only exposed to the ambient geofield.

statistical significance. A similar plot for each one of the two biological replicates composing the full dataset of 229 cells is found in SF3 Fig.

Such effects on F-actin dynamics could be promoted either indirectly or directly. Indirectly, a tensegrity response [55] of the cytoskeleton components resulting from MT disruption could modify F-actin organization in order to dynamically redistribute cellular mechanical stresses. As an alternative model, we cannot exclude a potential direct effect of the MF on F-actin fiber assembly. For that, we examined the effect of MF exposure on assays of actin polymerization in the absence of nocodazole, as detailed below in Subsection.

In summary, the weak MF by itself did not strongly affect cytoskeleton dynamics in cells at rest—as it is patent for the cells exposed to the MF but not to nocodazole in the third row of Fig 5A. In turn, after the nocodazole challenge, the weak MF interfered with the cells' capacity to recover their MT assembly, and altered their actin filament profile, notably yielding actin fibers that are less aligned yet higher in number.

### Weak magnetic fields increase actin polymerization

Cells exposed to the weak MF (but in the absence of nocodazole) show a small increase in actin polymerization, as measured by plate reader-based actin polymerization assays and depicted in Fig 5C. We performed four technical replicates of experiments with 2N (with N = {3,3,3,12}) individual wells seeded with cells following the same procedure outlined in steps 1–4 in Fig 4, using DMSO, not nocodazole, in step 3. N wells have been kept under the ambient geomagnetic field and N wells have been exposed to the MF. In these assays, fluorescence intensity is a proxy for actin polymerization levels. The fluorescence of pyrene actin (360 nm excitation and emission at 420 nm) was measured at baseline (3 min), after exposure to the ambient geomagnetic field or to the MF (480 $\mu$T for 20 min, data not shown in this manuscript but deposited in the Github repository [48]), and after incubation with polymerization buffer during 1 h in the absence or presence of the same weak MF. Stimulation with the MF in the absence of a polymerization buffer does not affect the fluorescence level (data not shown in this manuscript but deposited in the Github repository [48]). The plotted points represent the increment of fluorescence promoted by the 1 h-incubation with the polymerization buffer, when the baseline fluorescence is subtracted. For each technical replicate, we plot with a circle the average ratio of fluorescence between the MF-exposed cells and the cells kept under ambient conditions; the vertical bars indicate one standard deviation above and below the average value, obtained with compound error propagation. Again using compound error propagation statistics, we obtain as an average for the four technical replicates a fluorescence ratio of 1.08 $\pm$ 0.07. The data not being normally distributed, we ran a Mann-Whitney statistical test to check if the control and MF-exposed populations are significantly different. Using the fluorescence intensity, normalized to 100%, of the four control conditions, versus the control-normalized average values of fluorescence after MF exposure depicted with circles in Fig 5C, we find a p value of 0.0211 $\leq$ 0.05.

## Conclusion

We designed a water-cooled and incubator-compatible Helmholtz coil setup specifically to enable the study of magnetic fields effects on biological systems. The setup, whose full mechanical and electronic blueprints are freely available, allows for multi-directional exposure of cells to 1D magnetic fields by automated rotation; in turn, the water-cooling mechanism ensures temperature stability, thus eliminating potential artifacts in experiments.

The designed coils setup is used to show the effect of short-term exposure (1 or 3 h) to a weak magnetic field of 480 $\mu$T along the vertical direction on the cytoskeleton of a rat vascular smooth muscle cell line. We observe that such a magnetic field influences how microtubules and actin reassemble after depolymerization by nocodazole exposure; and increases actin polymerization, even in the absence of nocodazole, by a small but significant amount.

The fluorescence images in Fig 5A; the fluorescence images of the other 225 cells used to produce Fig 5B; and the code used to produce Figs 5B and 5C can be found at the Github repository [48]. The coil construction protocol described in this peer-reviewed article is published on protocols.io (dx.doi.org/10.17504/protocols.io.n2bvjneewgk5/v1) and is included for printing purposes as S1 File.

## Supporting information

**S1 File**.
(PDF)

**Supplementary Materials**.
(PDF)

## Acknowledgments

We thank Paul Weiss for his valuable feedback on the manuscript.

## Author contributions

**Conceptualization:** Abasalt Bahrami, Leonardo Y. Tanaka, Ricardo C. Massucatto, Francisco R. M. Laurindo, Clarice D. Aiello.

**Data curation:** Abasalt Bahrami, Leonardo Y. Tanaka, Ricardo C. Massucatto, Clarice D. Aiello, Francisco R. M. Laurindo.

**Formal analysis:** Abasalt Bahrami, Leonardo Y. Tanaka, Ricardo C. Massucatto, Francisco R. M. Laurindo, Clarice D. Aiello.

**Funding acquisition:** Francisco R. M. Laurindo, Clarice D. Aiello.

**Investigation:** Abasalt Bahrami, Leonardo Y. Tanaka, Ricardo C. Massucatto, Francisco R. M. Laurindo, Clarice D. Aiello.

**Methodology:** Abasalt Bahrami, Leonardo Y. Tanaka, Ricardo C. Massucatto, Francisco R. M. Laurindo, Clarice D. Aiello.

**Project administration:** Francisco R. M. Laurindo, Clarice D. Aiello.

**Resources:** Francisco R. M. Laurindo, Clarice D. Aiello.

**Software:** Clarice D. Aiello.

**Supervision:** Clarice D. Aiello, Francisco R. M. Laurindo.

**Validation:** Abasalt Bahrami, Leonardo Y. Tanaka, Ricardo C. Massucatto, Francisco R. M. Laurindo, Clarice D. Aiello.

**Visualization:** Abasalt Bahrami, Leonardo Y. Tanaka, Ricardo C. Massucatto, Clarice D. Aiello.

**Writing – original draft:** Abasalt Bahrami, Leonardo Y. Tanaka, Ricardo C. Massucatto, Francisco R. M. Laurindo, Clarice D. Aiello.

**Writing – review & editing:** Abasalt Bahrami, Leonardo Y. Tanaka, Ricardo C. Massucatto, Francisco R. M. Laurindo, Clarice D. Aiello.

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
