## [Decision Letter · Decision Letter 0]

18 Oct 2024

PONE-D-24-28285Automated 1D Helmholtz coil design for cell biology:

Weak magnetic fields alter cytoskeleton dynamicsPLOS ONE

Dear Dr. Aiello, Despite the many invitations, along a extended period of time, we only managed to secure one reviewer to your manuscript. Hence, to prevent you from waiting for a longer time for a feedback, we (exceptionally) decided to consider a single reviewer report the long time to give you a feedback on your manuscript.

Thank you for submitting your manuscript to PLOS ONE. After careful consideration, we feel that it has merit but does not fully meet PLOS ONE’s publication criteria as it currently stands. Therefore, we invite you to submit a revised version of the manuscript that addresses the points raised during the review process.

We look forward to receiving your revised manuscript.

Kind regards,

Alexandre Bonatto

Academic Editor

PLOS ONE

Journal Requirements:

2. Thank you for stating the following financial disclosure: This work was supported by: Young

Investigator-FAPESP (Funda¸c˜ao de Amparo `a Pesquisa do Estado de S˜ao Paulo) grant 2018/07230-5,

scholarship grants 2020/04280-1, 2022/16629-4, and 2023/05600-8 to L.Y.T.; CEPID (Centro de Pesquisa

Inova¸c˜ao e Difus˜ao)/FAPESP grant 2013/07937–8 to F.R.M.L.; Pioneer Science–Instituto D’Or grant, ONR

grant N00014-21-1-4011, NSF-Nano EAGER grant 2041158, and NSF-Bio EAGER grant 2114144 to C.D.A. 

3. Thank you for stating the following in the Acknowledgments Section of your manuscript: We thank Paul Weiss for his valuable feedback on the manuscript. This work was supported by: Young

Investigator-FAPESP (Funda¸c˜ao de Amparo `a Pesquisa do Estado de S˜ao Paulo) grant 2018/07230-5,

scholarship grants 2020/04280-1, 2022/16629-4, and 2023/05600-8 to L.Y.T.; CEPID (Centro de Pesquisa

Inova¸c˜ao e Difus˜ao)/FAPESP grant 2013/07937–8 to F.R.M.L.; Pioneer Science–Instituto D’Or grant, ONR

grant N00014-21-1-4011, NSF-Nano EAGER grant 2041158, and NSF-Bio EAGER grant 2114144 to C.D.A.

Please remove any funding-related text from the manuscript and let us know how you would like to update your Funding Statement. Currently, your Funding Statement reads as follows: This work was supported by: Young

Investigator-FAPESP (Funda¸c˜ao de Amparo `a Pesquisa do Estado de S˜ao Paulo) grant 2018/07230-5,

scholarship grants 2020/04280-1, 2022/16629-4, and 2023/05600-8 to L.Y.T.; CEPID (Centro de Pesquisa

Inova¸c˜ao e Difus˜ao)/FAPESP grant 2013/07937–8 to F.R.M.L.; Pioneer Science–Instituto D’Or grant, ONR

grant N00014-21-1-4011, NSF-Nano EAGER grant 2041158, and NSF-Bio EAGER grant 2114144 to C.D.A.

4. Please expand the acronym “NSF” (as indicated in your financial disclosure) so that it states the name of your funders in full.

7. We notice that your supplementary files are included in the manuscript file. Please remove them and upload them with the file type 'Supporting Information'. Please ensure that each Supporting Information file has a legend listed in the manuscript after the references list.

Additional Editor Comments:

Dear Dr. Aiello,

Despite the many invitations, along a extended period of time, we only managed to secure one reviewer to your manuscript. Hence, to prevent you from waiting for a longer time for a feedback, we (exceptionally) decided to consider a single reviewer report the long time to give you a feedback on your manuscript. 

Reviewers' comments:

Reviewer's Responses to Questions

**Comments to the Author**

1. Does the manuscript report a protocol which is of utility to the research community and adds value to the published literature?

Reviewer #1: Yes

2. Has the protocol been described in sufficient detail?

To answer this question, please click the link to protocols.io in the Materials and Methods section of the manuscript (if a link has been provided) or consult the step-by-step protocol in the Supporting Information files.

The step-by-step protocol should contain sufficient detail for another researcher to be able to reproduce all experiments and analyses.

Reviewer #1: Yes

3. Does the protocol describe a validated method?

Reviewer #1: Yes

4. If the manuscript contains new data, have the authors made this data fully available?

Reviewer #1: N/A

**5. Is the article presented in an intelligible fashion and written in standard English?**

Reviewer #1: Yes

6. Review Comments to the Author

Reviewer #1: ABSTRACT

Please recast here ‘’For more than fifty years, scientists have been gathering evidence of the biological impacts of weak magnetic fields.’’

Please recast here ‘’However, the lack of systematics in experimental studies has hampered research

progress on this subject.

Please recast here and don’t start a sentence with’’ To’’ ‘’To systematically quantify magnetic field effects in cell biology, it is crucial to produce fields that can be automatically adjusted and that are stable throughout an experiment’s duration, usually operating inside an incubator.’’

Introduction

Please recast here and don’t start a sentence with’’ As’’ ‘’As our environment becomes more polluted by electromagnetic fields from various sources, understanding the effects of being exposed to these small-magnitude fields is a timely endeavor.’’

Please recast here ‘’To unambiguously establish and understand the biological effects of weak MFs, it is crucial to perform well-controlled and reproducible experiments.’’

Please recast here ‘’Some experiments on bioelectromagnetics use permanent magnets, facing challenges such as fixed field strengths and non-uniformity.’’

7. PLOS authors have the option to publish the peer review history of their article (what does this mean?). If published, this will include your full peer review and any attached files.

Reviewer #1: **Yes: **Prof Charles Oluwaseun Adetunji

---

## [Author Response · Author response to Decision Letter 1]

7 Feb 2025

December 31, 2024 Response to reviewers

1. Does the manuscript report a protocol which is of utility to the research community

and adds value to the published literature?

Reviewer #1: Yes.

We thank the reviewer for acknowledging both the utility and the value to the community

of our protocol paper detailing the effect of weak magnetic fields on cellular cytoskeleton

configuration.

2. Has the protocol been described in sufficient detail?

Reviewer #1: Yes.

We thank the reviewer for acknowledging that our protocol, deposited in protocols.io,

contains sufficient detail for another researcher to be able to construct our incubatorfriendly

coils.

3. Does the protocol describe a validated method?

Reviewer #1: Yes.

We thank the reviewer for acknowledging that the protocol describes a validated method.

4. If the manuscript contains new data, have the authors made this data fully available?

Reviewer #1: N/A.

We respectfully point out that all our raw data and analyses are, in fact, publicly available

and deposited in Github.

5. Is the article presented in an intelligible fashion and written in standard English?

Reviewer #1: Yes.

We thank the reviewer for their assessment.

6. Review Comments to the Author

Reviewer #1: Abstract:

Please recast here: ‘For more than fifty years, scientists have been gathering evidence

of the biological impacts of weak magnetic fields.’

The sentence has been amended as follows: ‘Evidence of the biological impacts of weak

magnetic fields have been reported for more than fifty years.’

Please recast here: ‘However, the lack of systematics in experimental studies has

hampered research progress on this subject.’

The sentence has been amended as follows: ‘However, research progress on such effects

has been hampered by a lack of systematics in most experiments.’

Please recast here and don’t start a sentence with ‘To’: ‘To systematically quantify

magnetic field effects in cell biology, it is crucial to produce fields that can be

automatically adjusted and that are stable throughout an experiment’s duration, usually

operating inside an incubator.’

The sentence has been amended as follows: ‘Efforts to increase the systematics in

such cell biology experiments must include the capability of producing fields that can be

automatically adjusted and that are stable throughout an experiment’s duration, usually

operating inside an incubator.’

Reviewer #1: Introduction:

Please recast here and don’t start a sentence with ‘As’: ‘As our environment becomes

more polluted by electromagnetic fields from various sources, understanding the effects

of being exposed to these small-magnitude fields is a timely endeavor.’

The sentence has been amended as follows: ‘Understanding the effects of being exposed

to small-magnitude fields such as those produced by the low-level electromagnetic pollution

that permeates our environment is a timely endeavor.’

Please recast here: ‘To unambiguously establish and understand the biological effects

of weak MFs, it is crucial to perform well-controlled and reproducible experiments.’

The sentence has been amended as follows: ‘Well-controlled and reproducible experiments

are crucial to unambiguously establish and understand the biological effects of weak MFs.’

Please recast here: ‘Some experiments on bioelectromagnetics use permanent magnets,

facing challenges such as fixed field strengths and non-uniformity.’

The sentence has been amended as follows: ‘Certain bioelectromagnetics experiments

employ permanent magnets, which present challenges like fixed field strengths and uneven

magnetic fields.’

We do not find any additional comments in the report requiring further action. We sincerely

thank the reviewer for the feedback and are pleased that the latter sections of the manuscript

are acceptable as they are.

7. Do you want your identity to be public for this peer review?

Reviewer #1: Yes: Prof. Charles Oluwaseun Adetunji.

We wholeheartedly thank Prof. Adetunji for their time and interest, and appreciate their

choice to have their identity disclosed.

December 31, 2024 Additional journal requirements

1. Please ensure that your manuscript meets PLOS ONE’s style requirements.

We confirm that the manuscript meets the above style requirements.

3. Funding information should not appear in the Acknowledgments section or other areas

of your manuscript. We will only publish funding information present in the Funding

Statement section of the online submission form. Please remove any funding-related

text from the manuscript and let us know how you would like to update your Funding

Statement. Currently, your Funding Statement reads as follows: ‘This work was supported

by: Young Investigator-FAPESP (Fundação de Amparo à Pesquisa do Estado de São Paulo)

grant 2018/07230-5, scholarship grants 2020/04280-1, 2022/16629-4, and 2023/05600-8

to L.Y.T.; CEPID (Centro de Pesquisa Inovação e Difusão)/FAPESP grant 2013/07937–8 to

F.R.M.L.; Pioneer Science–Instituto D’Or grant, ONR grant N00014-21-1-4011, NSF-Nano

EAGER grant 2041158, and NSF-Bio EAGER grant 2114144 to C.D.A.’

2. Please state what role the funders took in the study.

4. Please expand the acronym ‘NSF’ (as indicated in your financial disclosure) so that it

states the name of your funders in full.

We have removed all funding information from the Acknowledgments section. The following

sentence has been added to the Funding Statement: ‘The funders had no role in study

design, data collection and analysis, decision to publish, or preparation of the manuscript.’

In the Funding Statement, ‘ONR’ now reads ‘Office of Naval Research’, and ‘NSF’ now reads

National Science Foundation.

In summary, the full and corrected Funding Statement now reads: ‘This work was supported

by: Young Investigator-FAPESP (Fundação de Amparo à Pesquisa do Estado de São Paulo)

grant 2018/07230-5, scholarship grants 2020/04280-1, 2022/16629-4, and 2023/05600-8

to L.Y.T.; CEPID (Centro de Pesquisa Inovação e Difusão)/FAPESP grant 2013/07937–8 to

F.R.M.L.; Pioneer Science–Instituto D’Or grant, ONR (Office of Naval Research) grant

N00014-21-1-4011, NSF (National Science Foundation)-Nano EAGER grant 2041158, and

NSF-Bio EAGER grant 2114144 to C.D.A. The funders had no role in study design, data

collection and analysis, decision to publish, or preparation of the manuscript.’

5. We note that you have included the phrase ‘data not shown’ in your manuscript.

Unfortunately, this does not meet our data sharing requirements. PLOS does not permit

references to inaccessible data. We require that authors provide all relevant data within

the paper, Supporting Information files, or in an acceptable, public repository. Please

add a citation to support this phrase or upload the data that corresponds with these

findings to a stable repository (such as Figshare or Dryad) and provide and URLs, DOIs, or

accession numbers that may be used to access these data.

The corresponding data has been uploaded to Github, and the two instances of the phrase

‘data not shown’ have been amended as follows, with corresponding hyperlinks: ‘The

fluorescence of pyrene actin (360 nm excitation and emission at 420 nm) was measured

at baseline (3 min), after exposure to the ambient geomagnetic field or to the MF (480 μT for

20 min, data not shown in this manuscript but deposited in Github), and after incubation with

polymerization buffer during 1 h in the absence or presence of the same weak MF. Stimulation

with the MF in the absence of a polymerization buffer does not affect the fluorescence level

(data not shown in this manuscript but deposited in Github).’

6. Please include captions for your Supporting Information files at the end of your

manuscript, and update any in-text citations to match accordingly.

We confirm that Supporting Information figures all have captions, and that in-text citations

are matched accordingly.

7. We notice that your supplementary files are included in the manuscript file. Please

remove them and upload them with the file type ’Supporting Information’. Please ensure

that each Supporting Information file has a legend listed in the manuscript after the

references list.

We separated the main manuscript from the Supporting Information file, and added a list of

the Supporting Information sections in the manuscript after the references list.

---

## [Editor Report · Decision Letter 1]

2 Mar 2025

Automated 1D Helmholtz coil design for cell biology:

Weak magnetic fields alter cytoskeleton dynamics

PONE-D-24-28285R1

Dear Dr. Aiello,

We’re pleased to inform you that your manuscript has been judged scientifically suitable for publication and will be formally accepted for publication once it meets all outstanding technical requirements.

Kind regards,

Baeckkyoung Sung, Ph.D.

Academic Editor

PLOS ONE

Additional Editor Comments (optional):

The authors have adequately responded to the reviewers’ comments.

---

## [Editor Report · Acceptance letter]

PONE-D-24-28285R1

PLOS ONE

Dear Dr. Aiello,

I'm pleased to inform you that your manuscript has been deemed suitable for publication in PLOS ONE. Congratulations! Your manuscript is now being handed over to our production team.

Kind regards,

on behalf of

Dr. Baeckkyoung Sung

Academic Editor

PLOS ONE